# Acid Catalyzed Stereocontrolled Ferrier-Type Glycosylation Assisted by Perfluorinated Solvent

**DOI:** 10.3390/molecules27217234

**Published:** 2022-10-25

**Authors:** Zhiqiang Lu, Yanzhi Li, Shaohua Xiang, Mengke Zuo, Yangxing Sun, Xingxing Jiang, Rongkai Jiao, Yinghong Wang, Yuqin Fu

**Affiliations:** 1College of Chemistry and Chemical Engineering and Henan Key Laboratory of Function-Oriented Porous Materials, Luoyang Normal University, Luoyang 471934, China; 2Hubei Key Laboratory of Natural Products Research and Development, Key Laboratory of Functional Yeast (China National Light Industry), College of Biological and Pharmaceutical Sciences, China Three Gorges University, Yichang 443002, China; 3Academy for Advanced Interdisciplinary Studies, Southern University of Science and Technology, Shenzhen 518055, China

**Keywords:** glycosylation, Ferrier rearrangement, perfluorinated solvent, high stereoselectivity, reusability

## Abstract

Described herein is the first application of perfluorinated solvent in the stereoselective formation of *O*-/*S*-glycosidic linkages that occurs via a Ferrier rearrangement of acetylated glycals. In this system, the weak interactions between perfluoro-*n*-hexane and substrates could augment the reactivity and stereocontrol. The initiation of transformation requires only an extremely low loading of resin-H^+^ and the mild conditions enable the accommodation of a broad spectrum of glycal donors and acceptors. The ‘green’ feature of this chemistry is demonstrated by low toxicity and easy recovery of the medium, as well as operational simplicity in product isolation.

## 1. Introduction

Facile and stereoselective construction of glycosidic linkages has always been one of the major focal points in the carbohydrate research community. Among these, the 2,3-unsaturated *O*-glycosides have attracted great attention because of their wide occurrence in bioactive molecules (Figure 1a) [1,2] and the potential for rapid functionalization [3,4]. Over the past several decades, various efficient methods have been established for forging such core scaffolds with a Ferrier rearrangement [5,6] that employs readily accessible glycals, and *O*-nucleophiles emerging as the most robust strategy [7,8,9,10]. Owing to the mild conditions and short reaction times, Lewis acids are the catalyst class of choice to promote this type of transformation [7,8,9,10], while Brønsted acids [7,8,9,10] and transition metal catalysts were also found to be effective [7,8,9,10,11,12]. Alternatively, a Ferrier-type *O*-glycosylation could be mediated by single-electron transfer reagents [13,14] via a radical pathway. These developments notwithstanding, a predominant *α*-selectivity in the formation of *O*-glycosidic linkages is normally dictated by multiple factors, including the conformation of glycal, anomeric effect, as well as the solvent effect in most cases [10,15,16] (Figure 1b).

In this context, palladium-catalyzed *O*-glycosylation with glycals as donors offers complementary and more programmable access by which excellent stereocontrol could be governed through the rational selection of the leaving group [17,18], ligand [19], or palladium source [20]. In this paradigm, tactics such as the addition of zinc reagent to render a softer acceptor [17,19], modification of glycal to activate the donor [21,22], or application of decarboxylative pathway to formally activate both reactants [23,24] are invoked to improve the performance of these reactions (Figure 1c). A review of these systems suggested that by incorporating a catalyst that could bring the donor and acceptor together through noncovalent interactions, the reaction might be catalytically mediated via a stereoselective manifold. Inspired by the recent advance in stereoselective *O*-glycosylation by means of bifunctional H-bond catalysis with *O*-acceptor [25], we envisioned devising a novel catalytic system to mimic this activation mode with other less-explored weak interactions.

Perfluorinated hydrocarbons displaying low chemical activity, low toxicity and low miscibility with common organic solvents have been recognized as a class of useful reaction mediums in various research fields [26,27], particularly in molecular-oxygen-involved aerobic oxidation reactions [28,29,30,31,32,33,34]. Wide application potential is also found in biphase catalysis by virtue of their unique physical properties [35,36]. Moreover, fluorous solvents could engage in diverse weak interactions such as π–π_F_, C–F···H hydrogen bond, C–F···C=O, and anion-π_F_, which play essential roles in the promotion of chemical transformations by enhancing reactivity and stereoselectivity as well as the design of functional materials [26,27,37,38,39,40,41]. In carbohydrate chemistry, it has been found that introducing a perfluorinated solvent could improve the reaction outcome [42,43,44,45]. These findings led us to postulate that the weak interactions stemming from perfluorinated solvent could be leveraged to improve the acid-catalyzed Ferrier-type glycosylation reaction (Figure 1d). On account of the weak acidic condition compared to traditional acid-catalyzed Ferrier rearrangement, the translation of this design into an effective process would further enable stereocontrol and broadens the substrate generality. Notably, the use of perfluorinated solvent could additionally ease the isolation of the glycoside products and the recovery of the reaction medium and assistor.

## 2. Results and Discussion

### 2.1. Optimization of Reaction Conditions

Based on these design criteria, the study on this stereocontrolled glycosylation commenced by employing tri-*O*-acetylated glucal **1a** as the donor, while ethanol **2a** serves as both the acceptor and solvent (Table 1). TFE (trifluoroethanol) was first attempted as the additive, which might promote glycosylation through acidic proton or/and other noncovalent weak interactions with **2a [46]**. Encouragingly, the *O*-glycosidic product **3a** was provided in 45% yield after 6 h at 100 °C (entry 1). The use of PFD (1H,1H,2H,2H-perfluoro-1-decanol) with a longer perfluorinated alkyl chain improved the yield to 55%, indicating the dominant role of the fluorine effect (entry 2). This speculation was further corroborated by the enhanced chemical yield when PFH (perfluoro-*n*-hexane) without an acidic proton was used as the catalyst (entry 3). Nonetheless, a significant decrease in conversion was observed when PFTEA (perfluoro-triethylamine) [47] was utilized, implying that the basic environment could retard the progress of this transformation (entry 4). It should be noted that high α-selectivity was detected for the generated *O*-glycosidic product for all these reactions (α:β > 20:1). Unsurprisingly, less than 10% yield and poor stereoselectivity (α:β = 5:1) was obtained in the absence of additive (entry 5). These results illustrated the positive effect of weak interactions on both efficiency and stereocontrol. As more complex glycosyl acceptors may not be accessed as easily and well-suited for use in solvent quantities, the reaction using stoichiometric glycosyl acceptors was evaluated in PFH due to the environmental friendliness and recyclability. However, under this set of conditions, only a trace amount of **3a** was detected (entry 6). Exogenous proton was introduced, and notably, 0.6 wt% of H^+^ type sulfonic resin (resin-H^+^) was sufficient to deliver a quantitative amount of glycosylated α-**3a** (entry 7). Meanwhile, when CH_2_Cl_2_ was used as the solvent, low yield (16%) and poor stereoselectivity (α:β =1.5:1) were delivered (entry 8). Similarly, the stereoselectivity was decreased (α:β = 7:1) when PFH was substituted by ethanol (entry 9), and no **3a** was obtained with less amount of PFH (10%) and *n*-hexane as a solvent, further affirming our hypothesis (entry 10). Other solvents were also screened, but no satisfactory results could be observed (entries 11–13). Lowering the temperature to 80 °C led to appreciable erosion of chemical yield (entry 14), whereas a prolonged reaction time of 14 h led again to a good yield (entry 15). A trace amount of **3a** was detected when the temperature was further decreased to 60 °C (entry 16). The absolute configuration of **3a** was determined by X-ray crystallographic analysis.

### 2.2. Substrate Scope

With the optimized conditions in hand, the substrate generality with respect to glycosyl acceptors was evaluated using glucal **1a** as the standard donor. As depicted in Figure 1a, various types of glycosyl acceptors, including alkyl, allyl, benzyl, and propargyl alcohols, could give the desired glycosidic products in excellent yield with high stereocontrol at the anomeric center (**3b**-**3o**, α:β > 20:1). It is noteworthy that sterically hindered (**3f** and **3j**) and structurally rigid (**3o**) alcohols that are unreactive reactants for conventional Ferrier rearrangement approaches could convert efficiently to respective *O*-glycosylation products. Subsequently, phenols with different substituents and substitution patterns were examined, and the glycosidic **3p**-**3ac** was synthesized smoothly (Figure 1b). Compared to aliphatic alcohol acceptors, the yields and stereoselectivities deteriorated in most cases, probably due to the strong background reaction catalyzed by an acidic hydroxyl group of phenols. Apart from *O*-nucleophiles, *S*-nucleophiles were also applicable for this reaction (Figure 1c). Although all the tested substrates reacted well with **1a** to give compounds **4a**-**4e** in good yields, the stereochemical outcome varied greatly. For instance, a 1:1 α:β mixture was detected for **4a** (from *n*-butylthiol) while **4b** (from *t*-butylthiol) was generated with an α:β ratio > 20:1. Likewise, thiophenol with electron-withdrawing group delivered *S*-glycosidic **4c** in poor stereocontrol while **4d** with an electron-donating group on thiophenol was obtained with α:β ratio of 10:1. When 2-methylbenzenethiol was utilized, the desired glycosylation product **4e** was formed in 75% yield with 6:1 α:β selectivity. Additionally, C-3 substitution products **4c’** and **4e’** were isolated alongside 6% and 8% yields, respectively. The absolute configurations of **3aa**, **3ab**, **4e,** and **4e’** were determined by X-ray crystallographic analysis, and those of other products in this scheme were assigned by analogy. Water also functioned well as an acceptor in the developed reaction, giving α-**5** an 87% yield.

Subsequently, the generality of this glycosylation method was studied with other types of glycal donors (Figure 2). Firstly, d-galactal **1b**, C-4 epimer of **1a** was employed, and the results were summarized in Figure 2a. A series of alcohols were examined, and these reactions invariably gave only **6a**-**6e** in excellent yields and α:β > 20:1. Phenols, thiols, and thiophenols were also applicable to afford **6f**-**6i** in good yields and stereoselectivities. As a C-3 epimer of **1a**, the combination of d-allal **1c** with selected glycosyl acceptors forged the corresponding products in more than 80% yield (**3a**, **3aa**, **4b,** and **4c**). Interestingly, remarkable α-selectivities were detected for all of these reactions, same with the case for glucal **1a** (Figure 2b). l-Rhamnal **1d** was also verified to be a competent donor for this transformation, and **7a**-**7d** was established with excellent outcomes (Figure 2c). However, when the pentose substrates were employed in this procedure, such as *D*-xylal **1****e** or d-arabinal **1****f** (a pair of C-3 epimers) as glycosyl donors, poor α:β ratios were observed for these reactions (Appendix A, **8a**-**8d**), indicating the direct significance of C-5 substitution in stereoinduction.

To demonstrate the practicality of the developed glycosylation strategy, the reactions of **1a** with an array of functional molecules as acceptors were investigated (Figure 3a). First, glycosylated product **9a** with a long alkyl chain was prepared in 90% chemical yield with α:β > 20:1, indicating the potential utility in lipidosome assembly. A fluorous tag containing long-chain linear perfluorocarbon was well tolerated to afford **9b** with the same level of outcome. Glycosylation with sugar alcohol delivered disaccharide **9c** in 80% yield with α:β selectivity of 12:1. When phenol derived from tetraphenylethylene with aggregation-induced emission attribute was reacted, **9d** could be generated in moderate yield with α:β = 9:1. Furthermore, the reaction operated smoothly on bioactive diosgenin to generate the C-O bond formation product **9e** with perfect stereochemical control.

A gram-scale reaction between **1a** and **2a** was also implemented under the standard conditions, in which the synthetic efficiency and stereocontrol observed for the small-scale reaction were perfectly preserved (Appendix A). Additionally, given the ease of isolation and good recyclability of organofluorine solvent, the recycling experiments were conducted to reinforce the utility of this strategy. After the completion of each reaction, the target product was easily isolated by phase separation, and the recovered reaction system (bottom phase) was reused successively. As summarized in Figure 3b, when ethanol **2a** was used to react with donor **1a**, the stereoselectivity (α:β > 20:1) was perfectly preserved, and the chemical yield was maintained at a good level (>70%) even after a repetition of this procedure for seven times. Similar results were obtained by using 3,4-dimethylphenol **2q** as a glycosyl acceptor for the recycling experiment.

## 3. Materials and Methods

The detailed procedure of the synthesis and characterization of the products are given in Appendix A.

## 4. Conclusions

In conclusion, an acid-catalyzed stereoselective Ferrier-type glycosylation assisted by perfluorinated solvent has been established. A wide range of glycal donors and glycosyl acceptors are well accommodated to provide structurally diverse *O*- and *S*-glycosylated linkages products in good efficiency for most cases. The utilization of perfluoro-*n*-hexane as the solvent improves the reaction conditions, increases the yield, and enhances the stereocontrol at the anomeric center. Notably, the turnover of this procedure is achieved with a minimal amount of resin-H^+^. Aside from experimental ease in isolating products, the use of low toxic and recyclable perfluorinated solvent highlights the environmental friendliness of the developed method.

## Data Availability

The data presented in this study are available in the Appendix A.

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
