# Peer review of "Acid Catalyzed Stereocontrolled Ferrier-Type Glycosylation Assisted by Perfluorinated Solvent"

_molecules, 2022, doi:10.3390/molecules27217234_

Round 1

Reviewer 1 Report

 The manuscript describes a method for stereoselective formation of O/S-glycosidic linkages via Ferrier rearrangement of acetylated glycals using perfluorinated solvent. The method has broad substrate generality and good to excellent alfa-selectivity for six carbon glycals. In addition, the perfluorinated solvent was recyclable. And the compounds' data was complete. This paper should be published in this journal after modification as described below.

1, In table 1 entry 6, there is almost no product without resin-H+ using PFH; while the normal solvents are not good for this reaction with resin-H+(entries 9-12). But the yield was increased significantly under resin-H+ as catalyst and PFH as solvent. It is better to explan why PFH can improve resin-H+ catalytic activity. And please reconsider the title and the sentance in conclusion "Notably, catalytic turnover is achieved with a minimal amount of resin-H+ due to the release of acetic acid in the reaction process."

2, In scheme 1 and scheme 2, the reactions generated the same products 3aa and 4c using different donor glucal 1a and allal 1c, but the stereoselectivity were different. it is better to give explanation.

3, There are some formatting errors in table 1 (superscript, italics, space) and references 4d.

4, For the PFH recycling experiments, does need to add 0.2mg resin-H+ for each cycle? I did not find this information in manuscript and SI. It is better to provide.

5, It is better to give references for known compounds. A few NMR spectra are not clean enough.

Reviewer 2 Report

In this manuscript, the authors reported a method for stereocontrolled Ferrier-type glycosylation mediated by per- 2 fluorinated solvent, where a perfluorinated solvent played a key role in stereoselective formation of O-/S-glycosidic linkages. Glycal donors and a wide range of acceptors were tested in the method, which provided structurally diverse O- and S-glycosylated products with good efficiency. An outstanding advantage of this method is that the low toxic perfluorinated solvent can be recycled and reused in the reaction system. This paper should be published in this journal after modification as described below.

1.       The structure of allal 1c and glucal 1a in TOC and Scheme 2 are drawn too similar, which will confuse readers.

2.       There are problems with the literature citation format throughout the article, and the authors need to refer to a template article for revision.

3.       In the introduction part and Figure 1a, the authors should give a brief introduction to the potential applications of the activity of these two bioactive molecules. The authors should also give a brief introduction to the “potential for rapid functionalization”.

4.       In footnote to Table 1, there should be a space between e and f. The authors called the perfluorinated solvent a catalyst, however they did not explain how the perfluorinated solvent catalyzes this reaction. From the analysis of the catalytic mechanism, it seems that the resin-H+ is more like a catalyst. The authors may avoid the word “catalysis” in the Table and combine them all in the “additive” column, such like “PFH (a catalytic amount)”.

5.       The authors said that ”Compared to aliphatic alcohol acceptors, the yields and stereose-lectivities deteriorated for most cases probably due to the strong background reaction catalyzed by acidic hydroxyl group of phenols”. Can the authors in detail what is the strong background reaction and how the strong background reaction affect the stereoselectivity? Is there any references?

6.       In conclusion, the authors concluded that “A wide range of glycal donors and glycosyl acceptors……”. However, I only saw one glycosyl acceptor (9c).

7.       Please carefully check reference 4a and 4d.

8.       The authors should provide proofs in SI to support the alpha/beta stereoselectivity for each compound and give explanations in detail how they calculated the alpha/beta ratios. Please carefully check the HRMS and the calculated values deviate greatly from the found values for some compounds (>30ppm). For example, for 4d,HRMS (ESI) m/z: calcd. for C17H20O5S Na+(M + Na)+359.0924, found 359.0625”, the deviation value is 0.0299 (>80ppm).

Reviewer 3 Report

this work by Zhi-Qiang Lu and co-workers presented Stereocontrolled Ferrier-type Glycosylation Mediated by Per- 2 fluorinated Solvent. Authors described a nice story around the per fluorinated solvent with an impressive amount of examples considering the substrate scope of both the glycosyl acceptors and donors. However, I am not convinced with their mechansim/role of the solvent (PFH) as the reaction mediator. Because there are significant amount of reports stating that the acids could catalyze this ferrier type alkylation including the Hexa fluoro iso proponol (see the following paper: Recent Developments in the Ferrier Rearrangement, doi.org/10.1002/ejoc.201300798). In my opinion, I don't think that PFH is mediating this reaction, the resin is playing a central role in mediating the reaction as seen with the condition 8 in the optimization table, as it deliverd 85% of the product in the absence of PFH. Which makes more sence to me as the acids chilate/active the acetyl group at 3rd position of the tri-acetyl glycal. Not mention that without the presence of the resin the authors could not be able to get the desired product as seen with the condition 6 of the optimization table. With this I strongly believe that the premise of this paper is misguided by the role of PFH. If that is the case there are many reports mediated by the acids (Lewis acids/bronsted acids) which makes the manuscript so week to be publishable. 

Compound 4c one H is missing in NMR. However, The NMR spectrum has 7 protons. Check the number of protons carefully for all the compounds.

Round 2

Reviewer 3 Report

See the attachment
